

# A product quality impacts of a mobile software product line: an empirical study

Luka Pavlič, Tina Beranič and Marjan Heričko

Faculty of Electrical Engineering and Computer Science, University of Maribor, Maribor, Maribor, Slovenia

## ABSTRACT

**Background:** The software product lines (SPL) enable development teams to fully address a systematic reuse of shared assets to deliver a family of similar software products. Mobile applications are an obvious candidate for employing an SPL approach. This paper presents our research outcomes, based on empirical data from an industry-level development project. Two development teams were confronted with the same functionalities set to be delivered through a family of native mobile applications for Android and iOS.

**Methods:** Empirical data was gathered before, during and after a year of full-time development. The data demonstrate the impact of a SPL approach by comparing the SPL and non-SPL multiple edition development. One family of products (Android apps) was developed using an SPL approach, while another (iOS apps), functionally the same, was developed without employing an SPL approach. The project generated a volume of raw and aggregated empirical data to support our research questions.

**Results:** The paper reports a positive impact of an SPL approach on product quality (internal and external) and feature output per week. As data shows, it also increases the delivery of functionalities (240% in 6 more editions), while investing the same amount of effort needed for a single-edition development. As a result of system-supported separation of development and production code, developers had a high confidence in further development. On the other hand, the second team delivered less new functionalities, only two new application editions, and lower software quality than the team that manages multi-edition development by employing an SPL approach.

# INTRODUCTION

Reuse is one of the fundamental disciplines in software engineering. It plays an important role in the development of new systems and in maintenance of existing ones. It is an important concept, especially when developing several simultaneous versions of the software. Usually, a software does not result in a single version or edition, specially tailored to certain customers. On the other hand, we are talking about diversity also when we have at the declarative level only one version of the software, i.e., the test version, the production version, etc. Even more, software can also be tailored to a specific set of hardware. Regardless of the domain, reuse plays a crucial role in successfully managing a set of similar software editions and versions. The software product lines (SPL) is an approach to reuse, employed in case where a family of products shares several common

Corresponding author
Luka Pavlič, luka.pavlic@um.si

functionalities. In addition to rare publicly available empirical evidence, software architects have to rely primarily on theoretical benefits and their lessons learned, while practicing the SPL approach to reuse.

In this paper we will present our research outcomes, done during 1 year of development of mobile applications for two mobile platforms, Android and iOS, sharing a common set of functionalities. Mobile applications are a part of a larger project which also included backend cloud solutions, a web portal, a media streaming server and tablet applications. Several editions of mobile applications, applications for the Android platform, were managed with the introduction and implementation of the Software Product Line (SPL) approach, while other set of applications, applications for the iOS platform, were managed with more traditional methods of reuse, e.g., branches in version management system, sharing the same codebase, but compiling it several times, using compiler directives, runtime checking, etc. Android and iOS development teams shared the same set of functionalities that had to be developed and they were given the same time to finish the implementation. This industry-based setup gave us the opportunity to explore and share interesting pre-, mid- and post-development empirical data, compiled to research observations on SPL approach implications.

The rest of the paper is organized as follows. The chapter "Related Work" gives a general overview on the state-of-the art approaches, emphasizing the SPL approach. Also, related work that deals with the same research is outlined. In the next chapter, the research method is discussed in detail. Research questions are presented. The project setup, methods, tool and SPL realization are presented. Chapter 4 outlines the domain in which the SPL approach was exercised. The chapter "Results" provides details on empirical data that is used to address research questions. The paper continues with "Discussion" providing an interpretation of the empirical data, focusing on comparing Android and iOS products and the velocity of the teams. The differences will be discussed together with long-term implications and both the positive and negative lessons learned. The paper finish with "Conclusion" that summarizes the most important findings of our research.

## RELATED WORK

The software product lines (SPL) approach to reuse in the software engineering area has been discussed and published for several years. It was introduced in the Software Engineering Institute (*Northrop, 2002*) and proved to be an adequate solution to reuse in special cases, when several software products share a majority of functionalities, while only a fraction of functionalities are edition-specific. The foundation book "Software Product Lines: Practices and Patterns" from *Clements & Northrop (2001)*. According to the original SPL idea, development efforts are directed towards developing core assets, while product development is a process of aligning core assets into final products. Management activities (including analysis and design) are shared among all products. *Northrop (2002)* also proposes several patterns and their variants, to be used for SPL-based development.

The SPL approach is explained in detail by the *Software Engineering Institute (2020a)*. They define software product lines as follows:

*A software product line (SPL) is a set of software-intensive systems that share a common, managed set of features satisfying the specific needs of a particular market segment or mission and that are developed from a common set of core assets in a prescribed way.*

As explained by Northrop (*Software Engineering Institute, 2020b*) SPL in addition to the existing mechanisms of reuse allow other levels of reuse—reuse at the level of larger software pieces. Besides reusing technical building blocks, these also include reusing procedures and rules, associated with the software. They include single analytics, planning, and management of software development. The SPL approach could be implemented when some of the following issues occur as a result of the complexity of the software (*Software Engineering Institute, 2020b*):

- we develop the same functionality for a variety of products and/or customers,
- the same change should be made in a number of different software products,
- the same functionality should behave differently depending on the final product,
- certain functionality can no longer be maintained, and so the customer has to move to a newer version of the software,
- we cannot estimate the cost of transferring certain features to different software,
- certain basic infrastructure changes lead to unpredictable behavior of dependent products,
- the majority of effort is put into maintenance, and not the development of new functionalities.

The additional costs of the SPL approach are also clearly stated: architecture, building blocks and individual tests should include the possibility of variability, while business plans must be made for multiple products, not just one. The long term claimed contributions of SPL are as follows (*Software Engineering Institute, 2020b*): up to 10× improved productivity, up to 10× improved quality, joint development costs reduced by up to 60%, shortened time-to-market by up to 98% and the possibility of moving to new markets is measured in months, not in years.

SPL positive effects could, however, only have been observed if SPL approaches were used correctly and in appropriate software development projects. *Muthig et al. (2004)* lists several possibilities of misusing the SPL approach. These include (*Muthig et al., 2004*) general purpose reuse, using classical reuse techniques such as component-based reuse, having configurable software behavior and managing versions of the same software. According to many authors, an important aspect in the SPL approach is managing variabilities. *Cavalcanti, Machado & Anselmo (2013)* define the SPL-approach as a tool to effectively cope with variabilities. The authors address three types of variabilities:

- Functionality presence: If the functionality is present in all the lines and in all with the same realization, such functionality may be realized in the most general common building block.

- The lack of functionality: the functionality is not present in particular lines. In the case that the functionality is required in only one line, the functionality may be realized in the line itself, otherwise it is necessary to introduce a specific building block.

- A different realization: the functionality is available, but the realization will be different in different product lines. A different realization can be realized in the line, unless the same feature can be found in multiple lines—in this case, it is reasonable to introduce a new building block, which is a specialization of the existing one.

*Clements & Bachmann (2005)* explains that the technical realization of variabilities is based on already established and well-known concepts in software engineering, including component level reuse, design pattern employment, developing plug-ins, using parameterization, configuration with deployment descriptors and others.

SPL approach specifics for modern mobile platforms and the specifics of mobile application development have been little published so far. However, some authors have published papers in this area. *Muthig et al. (2004)* published a report on The Go Phone Case Study. It is an in-depth report on the practical proof-of-concept development of mobile applications for several platforms; including SPL-specific analysis and design.

*Usman, Iqbal & Khan (2017)* shows two case studies employing SPL approach as well. Authors base their case studies on problem of developing and maintaining multiple native variants of mobile applications to support different mobile operating systems, devices and varying application functional requirements. Their approach includes their own tool (Moppet) to automates their approach of model-driven generating mobile applications. Authors deal with three types of variations in mobile applications: variation due to operation systems and their versions, software and hardware capabilities of mobile devices, and functionalities offered by the mobile application.

Similarly, *Marinho et al. (2013)* discusses the use of an SPL approach in mobile development for several hardware and context situations. They presented Android applications that resulted from the MobiLine project as well as the approach used to build them. The SPL employment resulted in 57 applications that share 100% common mobility-related functionalities, 87% common message exchange-related functionalities and 8% context-related functionalities. For modeling and building applications they proposed and used their MobiLine development approach. The authors only reported on the lessons learned from technical and project management areas.

*Alves, Camara & Alves (2008)* presented successful SPL employment in the area of mobile games. They observed an interesting fact—the SPL approach has been used several times in the past, but they did not name it SPL. It emerged naturally. At the same time, they reveal the need for standards or standard approaches in terms of establishing a common SPL architecture—which would simplify SPL-based development dramatically.

*Quinton et al. (2011)* reported on the problem of addressing variabilities, while designing mobile applications. The SPL approach is defined by two independent dimensions: mobile device variety and mobile platforms. Their model-driven approach, supported by the Applitude tool, enables the creation of Android, iOS and Windows Phone mobile applications, while considering variabilities. The variabilities are addressed

with merging core assets, while the platform dimension is supported by metamodel in Applitude.

*Dürschmid, Trapp & Döllner (2017)* acknowledges the SPL approach in paper "Towards architectural styles for Android app software product lines" to be complex in Android development due to inflexible operating system framework. This is why they propose several techniques to achieve appropriate SPL architectures. They include activity extensions, activity connectors, dynamic preference entries, decoupled definition of domain-specific behavior via configuration files, feature model using Android resources. Using their approach, authors manage to demonstrate the benefits via 90% of code reuse in their case study. As we will demonstrate in this paper, our real-life production project goes even beyond techniques mentioned before (we also employ design patterns etc.) and demonstrate benefits not only via code reuse, but also via increased productivity and code quality.

As demonstrated in systematic literature reviews by *El-Sharkawy, Yamagishi-Eichler & Schmid (2019)* and *Chacón-Luna et al. (2020)*, SPL has gained momentum lately and the interest in empirical data on SPL has increased (*Chacón-Luna et al., 2020*). In addition to this, Software Product Lines Conference (*SPLC, 2020*) is organized annually, regularly disseminating the progress of this research domain. We see our papers' role as an important report of industry-scale development with shoulder-to-shoulder comparison of parallel development of the same application—one team with, and one team without SPL approach.

Another indicator of SPL approach gaining popularity in the mobile development is also the fact, that in 2019, Android API and Android Studio have a full support for "product flavors", which is Google's term for SPL—creating different variants of an app (*Android Developers, 2020*).

Product flavors allow the developers to specify different features and device requirements as well as use specific source sets for each flavor, while still using shared code and assets where possible. Each build variant represents a different version of an app built from a single project (*Android Developers, 2020*). When building the app only the source sets relevant to the selected build variant are included in the resulting end-user package, while all the others are left out. The Googles' approach to SPL is similar to the approach, presented in this paper. However, our approach, as demonstrated later in the paper, is based on projects, libraries, proven design patterns and avoids compile-time separation of product lines. In addition, even app resources (such as graphics, multilanguage translations etc.) are addressed by variabilities management, presented in this paper.

## RESEARCH METHOD

24alife is the ecosystem of information solutions, oriented towards an increasing quality of life. This multidisciplinary project includes medical, sports, psychological and nutritional aspects as well as the combined view of an individual through targeted support in detecting, monitoring and eliminating the negative effects of stress. It is intended for

individuals in strengthening and upgrading health reserves and, as such, focuses on a healthy lifestyle.

Mobile applications (Android, iOS) are used as a supplement to the primary, web-based, user interface. Their main role is to track and guide sports activities (such as jogging, cycling, fitness training, etc.), to do daily measurements (heart rate, weight, blood pressure etc.), motivate users, offer guidance for portal-created personal programs, etc. In addition to simple, practical suggestions, analysis and progress indications, they are also tasked with exchanging data with the cloud. There are several publicly available editions of mobile application, such as free and payable version. In addition to this, some editions are available only to project teams and partners and special customers. At the moment, all components support 6 languages and the imperial and metric system.

The 24alife project includes two mobile development teams—Android and iOS. Application families are developed from single requirement-design body in separate development teams, resulting in native applications for Android and iOS.

Two agile development teams of the same sizes, one for Android another for iOS, were given with the same set of requirements during the development. Both teams were supported by the same graphical designer. Product owner was also the same person for both teams. So, effectively, the goal was that they deliver two identical native mobile applications with the same set of functionalities and the same look. The only planned difference was expected behavior, aligned with iOS and Android design guidelines.

Presented circumstances enabled us to design a research which would quantify development teams design decisions on how to manage product families. The empirical data results from a process, illustrated in Fig. 1. To verify whether development teams, separately working on mobile applications for Android and iOS are comparable, a pre-development survey was executed, capturing the developers' experience and perceived knowledge. We designed our questionnaire based on the practices set forth (*Chen et al., 2018*). We asked them to enter their perceived level of knowledge of programing languages and provide a number of years for their professional experience. Since the knowledge self-assessment can be biased and subjective, the years of experience criterion was added in order to objectify participant's experiences.

The development team for the iOS application managed their multiple editions of mobile applications manually. The Android development team did the same during the weeks 1–33. From week 34, the Android development team switched to an SPL-based approach. In week 55, our research ended. During development weeks 1–55, we continuously measured internal quality metrics and source code size. It resulted in empirical data, used to verify if we can compare iOS and Android source code in the first place. Secondly, if and what is the impact of week 33s' decision on Android source code and further delivery of functionalities. We measured external quality via analyzing test reposts, generated by the quality assurance team. In addition to measuring source code size, application size was measured by counting functionalities, offered to users by application editions, which is also the case in function point analysis, which is one of the standard metrics for determining the size of the software product (*Albrecht, 1979*). The source code size (LOC—Lines of Code metric) was continuously monitored during

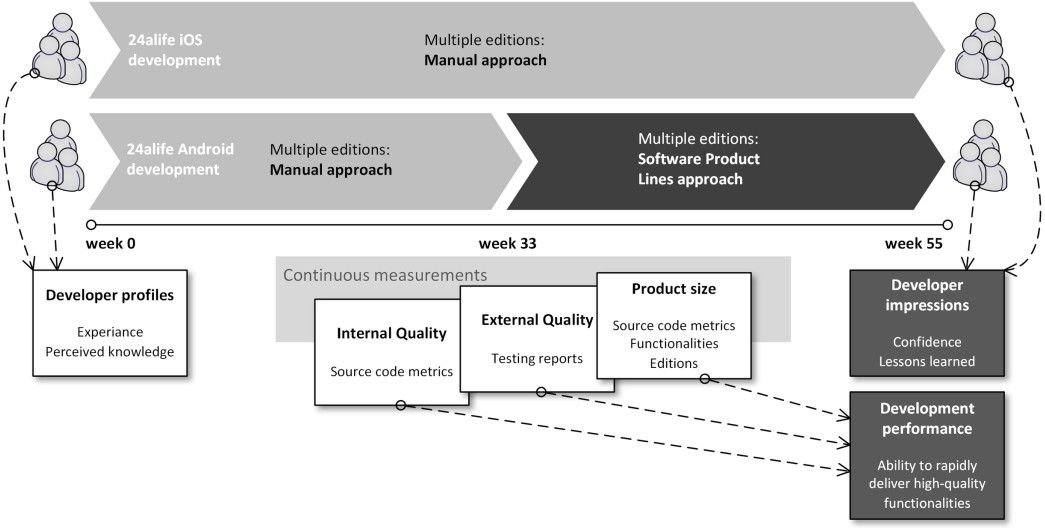

**Figure 1 Gathering research empirical data during the development process.**

the development not only to provide evidence on ability to compare iOS and Android application, but also to support possible differences in delivered functionalities in the second half of the development. In the first half of the development project, LOC metric was primarily used to verify if we can compare products (development effort and delivered functionalities were the same in both development teams). In the second half of the project, LOC was primarily a control variable to see if changes in delivered functionalities was a result of possible changes in particular team's effort. Deviations in size-based metrics would signal that the introduction of the SPL approach was not the only change in the development process.

Post-development gathering of empirical data consists of a questionnaire to capture developers' perception of their decision to manage multiple editions. It also included gathering and interpreting empirical metrics from the configuration management system, bug tracking system and release logs. Post-development data gathering aims is to show if introducing the SPL approach returned the investment in terms of changed delivery times and possible quality changes.

Based on the presented research method and compiled empirical data, we would like to conclude while answering the research questions below:

- **RQ1**: Is the source code size comparable between Android and iOS applications, while having the same set of requirements?
- **RQ2**: What are the impacts of introducing a software product lines (SPL) approach to a mobile development?

  - **RQ2.1**: What are the impacts of the SPL approach to application growth?
  - **RQ2.2**: What are the impacts of the SPL approach to application quality?

- **RQ2.3**: What are the impacts of the SPL approach to developers' confidence in frequent releases?

The presented set of research questions was carefully selected to check:

a) Whether we can compare Android and iOS applications on source code level, while having the same set of functionalities and investing the same effort amount;
b) Internal and external quality of both applications with respect to selected multiple editions approach;
c) Delivered application editions and core, optional and alternative functionalities with respect to selected multiple editions approach;
d) Developers perception on the selected multiple editions approach.

In addition to two questionnaires (pre- and post-development questionnaire), our research relies on software metrics, that support answers on research questions:

a) Source code size and class-based metrics (Lines of Code, Logical Lines of Code, Number of Classes) in combination with product size metrics (Number of Editions, Number of Functionalities) support RQ1;
b) Source code size-based metrics support RQ2.1 as an implicit control of invested effort to the development;
c) Product size metrics support RQ2.1;
d) Source code internal quality metrics (Code to Comment Ratio, Logic Density, Code Structure Modularity) support RQ2.2;
e) Product quality metrics (Number of reported errors, imperfections and inconsistencies) support RQ2.2;

## OUR APPROACH TOWARDS SEVERAL MOBILE APPLICATION EDITIONS

The 24alife project (see "Research Method") includes two mobile development teams (Android and iOS), driven by the same requirement-design body and resulting in native applications for Android and iOS.

The mobile development teams were combined in week 1. Weeks 1 to 3 were dedicated for preparations, reviewing the initial product backlog, early prototyping, deciding on architecture design, preparing user interface wireframes and designs. Functionality-driven development, i.e., the first iteration, started in week 4. The development practices and process mechanics were organized according to Scrum development method (*Sutherland & Schwaber, 2014*) by practicing planning, daily and demo meetings, retrospectives, honest sprint commitment via planning poker and others. Both development teams (iOS and Android) consisted of three experienced developers. Both teams share the same scrum master, product owner, graphics designer and quality assurance team.

In order to manage several editions efficiently, during requirements gathering, designing and testing, functionalities were collected in a multi-dimensional table. Functionalities were not only listed, but also described in terms of which edition functionality was available and if and what specialities were required for a particular functionality in a particular edition. This is how development team ended with functionalities written in several categories: common (all editions), optional (only in selected editions) and alternative (edition-specific implementation of the same functionality). Such approach enabled development teams to reuse common requirements, design and testing, which is also one of the foundations in order to establish SPL approach (*Software Engineering Institute, 2020b*).

Both teams initially managed multiple editions of their applications (daily built, test version) in manual manner. The breaking point (see Fig. 1) was at week 33 with a clear clients' demand do provide application in freely available ("Free") and payable editions ("Pro"). Since the iOS development team was confident, that their established techniques to manage several editions will continue to prove itself to be adequate, they did not change the design of their application (preserving a single development project with compiler directives and run-time switches, finally manually building the desired edition while providing an appropriate set of configurations). iOS development teams' approach is the best described as "ifdef hell", reported by several authors, e.g., (*Feigenspan et al., 2013*).

On the other hand, Android development team decided to invest extra effort to prepare the production-ready appropriate SPL architecture, libraries and projects.

A Set of core Android mobile development assets and available product lines is presented in Fig. 2. Figure 2 shows available assets (components implemented as Android libraries), from which 7+1(Core Module) are fully functional Android applications, combined from other assets. Figure 2 does not capture variabilities in functionalities (which can be seen in Table 1), rather it shows available components. A set of functionalities is present in particular application edition by appropriate library in edition. Functionality absence in achieved by not including the library. The alternative implementation is achieved by including library and overriding (a part) of its implementation by employing appropriate design pattern. A code-generation approach to introduce common, optional or alternative functionalities was not employed in the presented SPL architecture. No additional source code was automatically generated in order to support SPL approach.

The "24alife Core Module" is the Android library (at the same time a fully running Android application). It realizes functionalities that are common to all editions (product lines). The "Core Module" also contains common architecture of mobile applications and the variability points of additional functionalities (see Fig. 3). Applications within the product line are built into the final products with standard build tools, available in Android Studio. Core product assets (Android libraries) contain, in addition to the source code, also XML-written user interfaces, graphics, multilingual messages, dependant libraries, static content, etc. The library structure is shown in Fig. 3.

Android development team implemented variabilities using industry-proven best practices and approaches. These include the use of object-oriented design, proven design

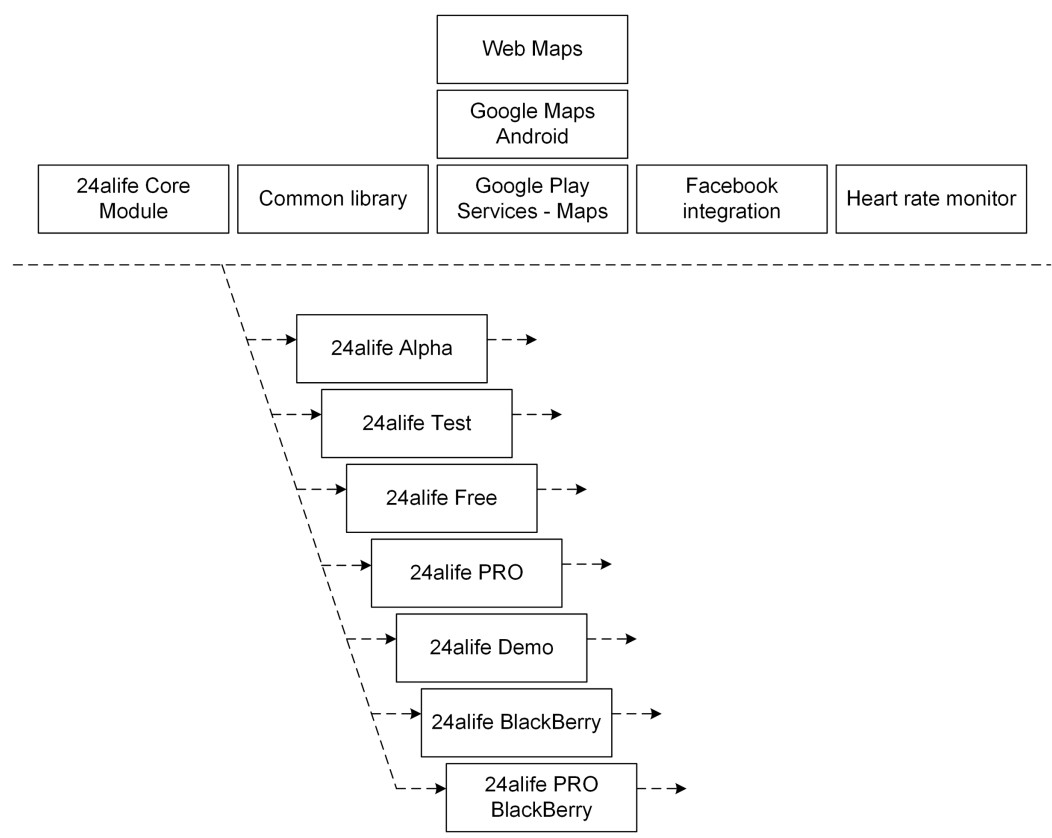

**Figure 2 Components (Android libriaries) in Android software product line.**

**Table 1 Final editions compared: functionality-based differences.**

| Edition | Based on | Base f. | Optional f. | Alternative f. | F. count (B+O) | F.diff (O+A) | Diff (%) |
|---|---|---|---|---|---|---|---|
| Pro | Free | 45 | 8 | 2 | 53 | 10 | 19 |
| Free | Core | 45 | 0 | 3 | 45 | 3 | 7 |
| Alpha | Core | 45 | 9 | 5 | 54 | 14 | 26 |
| Test | Pro | 53 | 0 | 1 | 53 | 1 | 2 |
| Demo | Free | 45 | 8 | 2 | 53 | 10 | 19 |
| BB Pro | Core | 45 | 0 | 3 | 45 | 3 | 7 |
| BB Free | Free | 45 | 0 | 3 | 45 | 3 | 7 |

patterns, extensions, and component parameterization. Design patterns are used heavily, especially: factory, abstract factory, factory method, bridge, bean, adapter and others (*Gamma et al., 1998*). As shown in Fig. 3, Android developers created a common SPL architecture as a set of concrete and abstract classes. Their primary role is to handle the orchestration of newly introduced functionalities from other libraries via prepared extension points.

Component-based development is heavily used in the presented SPL realization. All components are managed (created, initiated, cleaned etc.) via the Component Manager

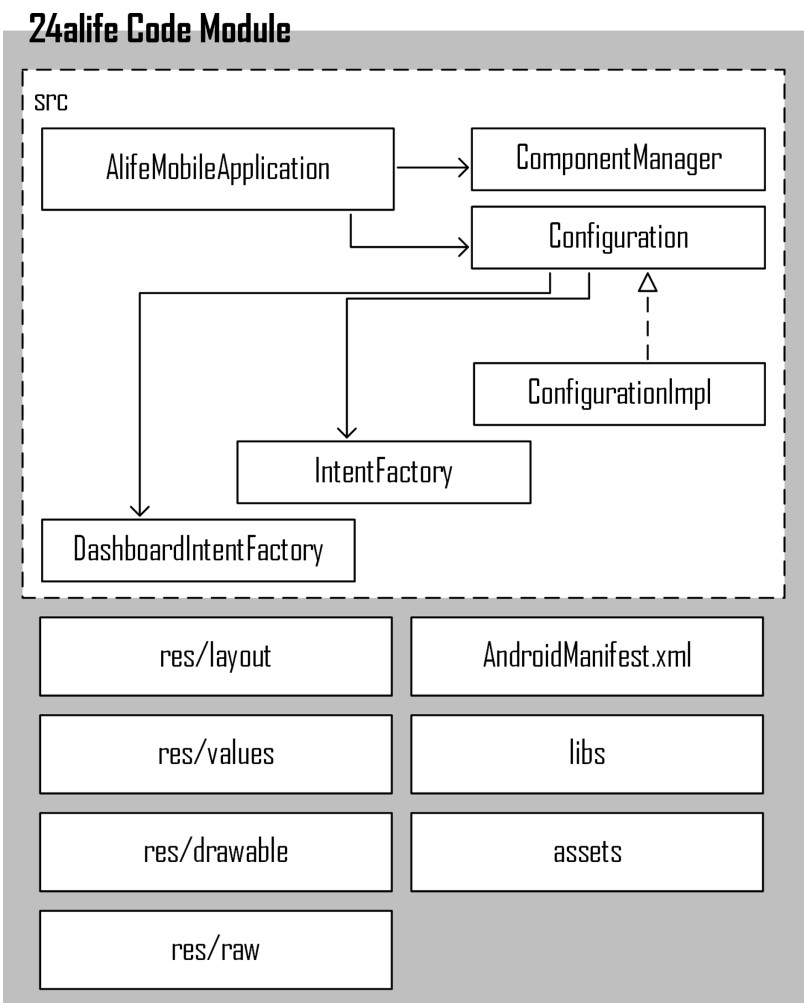

**Figure 3  SPL architecture as a part of Core Module.**

class. The Component contains the implementation of predefined interfaces and specific interfaces with their implementation as well. Operations, such as preparing the database structure, upgrading the database, exchanging data with the cloud (synchronizing) and others are all part of a component. This is how certain editions (product lines) without a certain component, will not even have database structure for unsupported functionalities. The component manager is also responsible for component-to-component communication (The Component Manager would provide a reference to a component in the same product line).

The Configuration class constitutes another concept in the presented architecture. Configuration, provided by the Core Module, includes general purpose configurations, common to all product lines. In a particular edition (the product line), developers would typically extend the Configuration class in order to do necessary adoptions or overriding (e.g., introduce new operations to be run when the mobile application starts, add new synchronization points, turn off some components, set data for accessing a particular cloud—whether it be production or test deployment).

This is how the inclusion of a specialized functionality in the individual line is, technically, achieved using several approaches:

- preparation of the extension point in the core module (in terms of components or inside components using factory method or abstract factory design patterns),
- using inheritance and adding new methods and/or method invocations in the product line,
- using an abstract factory pattern, which combines the functionality of the new line and its own user interface.

The exclusion of unwanted features is achieved mainly through inheritance and the exclusion of unwanted features (such as not downloading programs for free products), as well as with the parameterization of the basic building blocks. However, in most cases, exclusion was not necessary, since product line-specific functionalities were introduced in the product lines themselves (e.g., 24alife PRO library).

Changing behavior (e.g., the demo edition expires 1 month after construction), are achieved also with inheritance and/or by employing appropriate design patterns, such as: a bridge, factory method, or builder.

Based on presented technical SPL architecture, 7 application editions were managed. Table 1 summarizes functionality-based differences between them. For example, final "Alpha" edition is based on "Core Module", having 45 common functionalities, 9 additional functionalities are implemented only in "Alpha". Which results in total of 54 functionalities. In addition to this, 5 functionalities in "Alpha" are adopted to different behavior. Which resulted in 14 functionalities implementation for "Alpha" (optional + alternative). That is 26% difference with base edition, which is "Core Module" in case of "Alpha". Differences in terms of functionalities for other editions are demonstrated in Table 1.

# RESULTS

In order to answer the research questions, we will outline the most important empirical research data. This includes an in-depth analysis of version control logs, source code metrics for comparative quantitative and qualitative data, querying bug-management logs and production logs and analyzing developer questionnaires. Several tools were used in order to capture, prepare and aggregate raw data, including Subversion CLI, CLOC tool, Javancss, XClarify, ProjectCodeMeter and shell scripts for batch processing and data aggregation.

## Developer profiles

Based on the pre-development questionnaire, the mobile developer's profile was as follows:

- Android developers average perceived developments skills: 2.3/5,
- Android developers average perceived Android Studio skills: 4/5,
- iOS developers average perceived developments skills: 2/5,
- iOS developers average perceived XCode skills: 3/5,

- Android developers average experience in developing mobile applications: 4 years,
- iOS developers average experience in developing mobile applications: 3.3 years.

## Edition and functionality growth

During the research, presented in this paper, we observed the development of 55 weeks (see Fig. 1). During this time, the final version of the released mobile application is 3.0. Versions 1.0, 1.1, and 2.0 were released in a two-edition manner (free, pro) while version 2.1 and later were required in several editions (including demo, alpha, test, blackberry-optimised free and blackberry-optimised pro). Please see Table 2 for the growth in terms of new functionalities. Please also see Table 1 for a complete functionality count per particular edition.

Besides functionality growth, edition growth is also an important data regarding our research. Figure 4 shows the number of released products during project time (editions and versions combined) for iOS and Android. Since Android developers used an SPL approach, the chart in Fig. 4 provides additional insight into core SPL asset numbers over time. The chart clearly shows only version-based growth until week 41. After week 33, the Android application becomes a core asset, and the iOS application stays monolithic. Please note that Fig. 4 does not capture internally available editions. The chart includes 7 different Android editions across several versions and 2 iOS editions across several versions.

## Source code size

The product's quantitative metrics are as follows. One of the commonly used metrics is LOC (Lines Of Code). Figure 5 displays how LOC (without blanks and comments) changed over time for Android and iOS products. In both, an important segment of code is done in XML (examples would be user interfaces, navigation rules, animations, etc.) This is why we show this distribution also in Fig. 5. In the case of Android, the chart captures LOC for the entire Android projects (all editions, all core SPL assets). The chart includes the iOS initial application as well as the Free and Pro iOS editions. The internally available experimental iOS HRV/Corporate is not included, since it is a separate branch on its own. Please note, that week 4 was the first official iteration, while weeks 1–3 were preparations and prototyping. This is why the subversion repository is starting the codebase with week 4 and charts in Figs. 5 and 6 also start at week 4.

While LOC can give concrete insight into software size, we included the number of classes for both development projects in Fig. 6 in order to give more accurate insight into code distribution. The measured codebase is the same as in the chart in Fig. 5—all Android editions and versions with core SPL assets, iOS initial, Free and Pro editions.

We measured LOC values for both iOS and Android with the same tool (cloc), so that the same rules would apply, thus making the results comparable. We also used the ProjectCodeMeter tool to measure the final state of subversion repository for both projects as well as the final single-edition project for both projects. This also makes the results directly comparable.

**Table 2  Functionality count, implemented in iOS and Android mobile applications.**

| Public version | Included functionalities |
|---|---|
| 1.0 | 16 initial, core functionalities (e.g., login with portal-created account, data synchronization, settings, GPS-based activity tracking, enter medical measurements, view exercise videos, sports tests etc.) |
| 1.1 | 6 new functionalities (e.g., notifications, history management), general improvements |
| 2.0 | 9 new functionalities (e.g., support for guided, goal-based programs), general improvements |
| 2.1 | 2 new functionalities (including VO2max support) |
| 2.2 | 2 new basic functionalities (e.g., start-in-10) and 4 new Pro functionalities (e.g., progress analysis), general improvements |
| 2.3 | 4 new functionalities (e.g., Facebook integration), general improvements |
| 3.0 | 6 new basic functionalities (e.g., manual activity import) and 4 new Pro functionalities (e.g., goal-driven activities), general improvements |

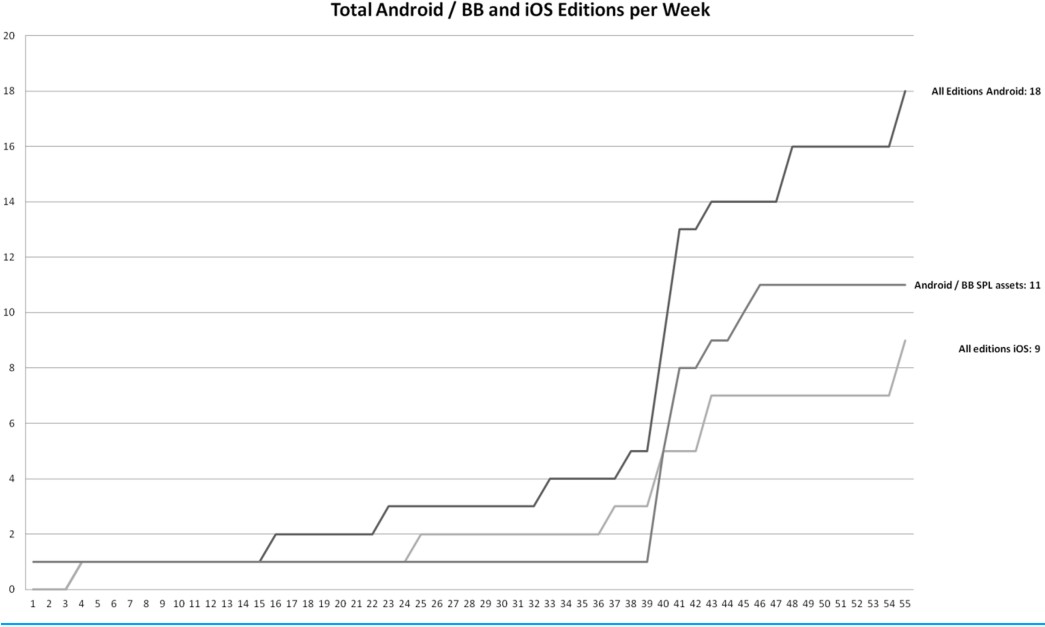

**Figure 4  Android and iOS mobile application editions available during the project.**

## Source code quality

In addition to size-oriented metrics, we also applied quality-oriented source code metrics. Since we are interested in internal quality change, measuring mid- and final-version of source code is sufficient. The results that the ProjectCodeMeter tool produces are summarized in Table 3. Code to Comment ratio (CCR) grabs logical LOC per one comment line. Code Structure Modularity (CSM) aggregates modularity of the source code (value: <100—low modularity, >100 fragmented code). Logic Density (LD) assesses number of lines to capture a certain logic. As Table 3 shows, CCR value stays the same (19) till the end of the project for the iOS application. On the other hand, we can see significant improvement (from 9 to 15) in week 55 in the case of the Android application. Code structure (CSM metric) is reduced in case the of iOS application source code (152 to 143).

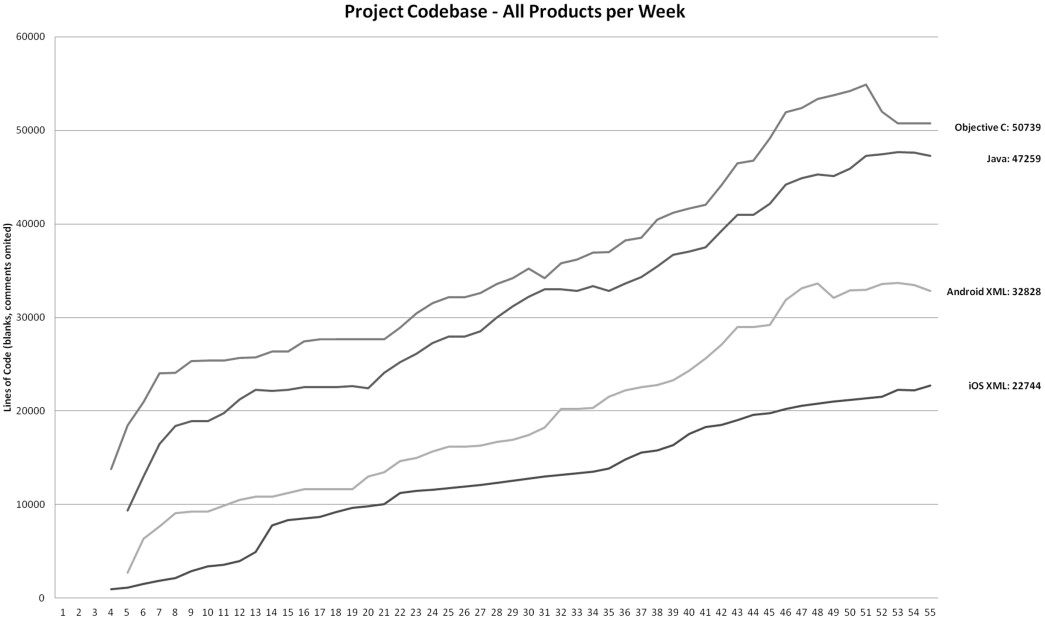

**Figure 5 Code size (LOC, no comments or blanks) during the project.**

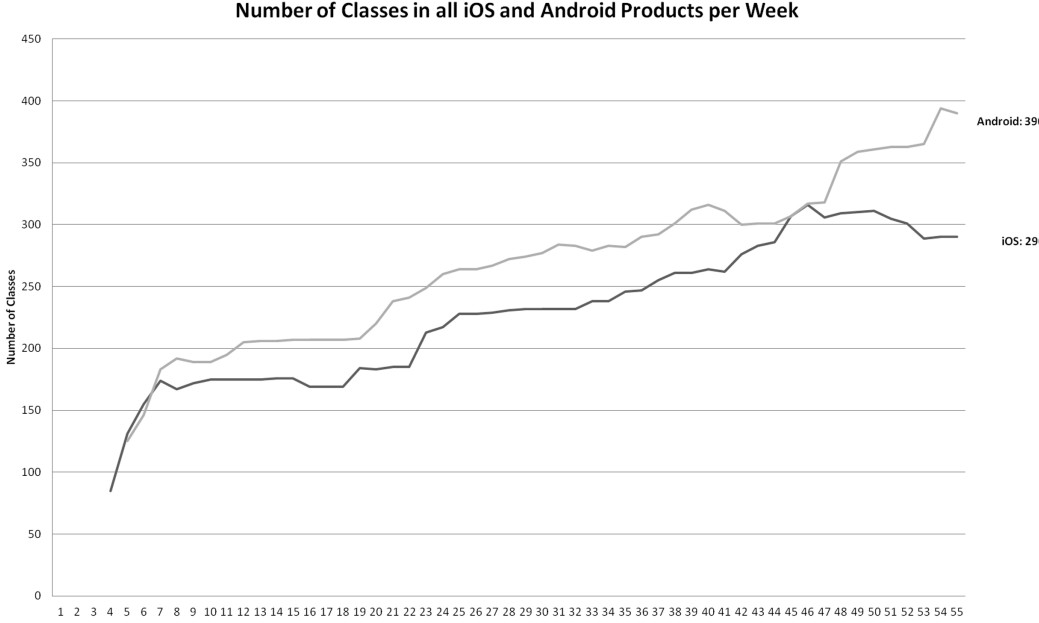

**Figure 6 Classes count during the project.**

**Table 3 Comparable final project metrics.**

|      | iOS (week 37) | iOS final | Android (week 33) | Android final |
|------|---------------|-----------|-------------------|---------------|
| CCR  | 19            | 19        | 9                 | 15            |
| CSM  | 152           | 143       | 145               | 153           |
| LD   | 77            | 81        | 45                | 66            |

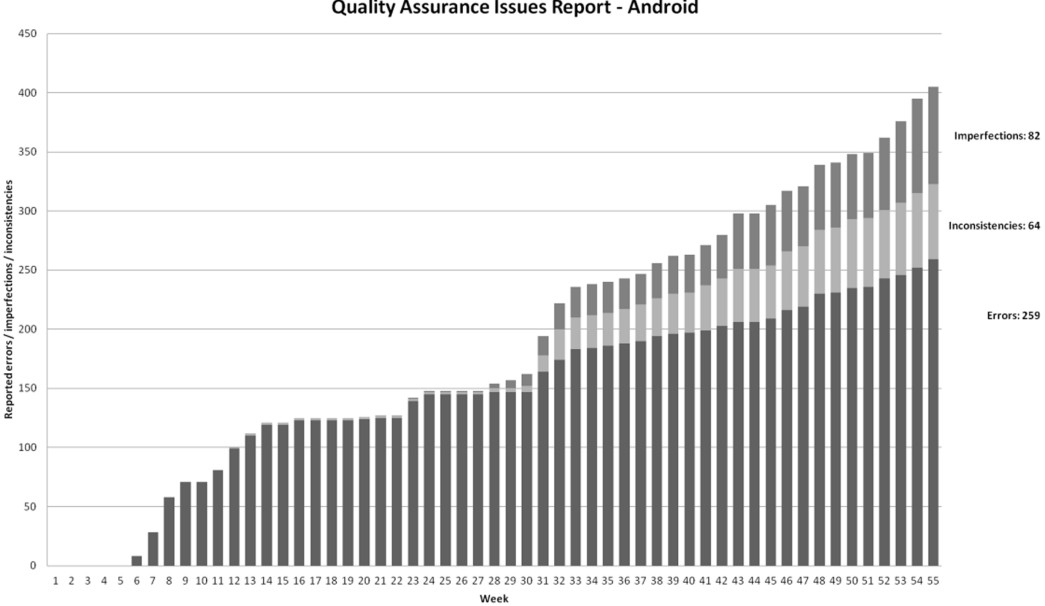

**Figure 7 Cumulative reported failures during time spent on Android projects.**

It is, however, the opposite in the case of Android application source code—CSM value is increased (145 to 153). Logic density (metric LD) is increased in both cases: Android application from 45 to 66 and iOS application from 77 to 88.

## Product quality

During the research, we also measured the product's external quality. The quantity of reported failures was used as a comparable measure. The quality assurance team was using the following classification of failures (based on IEEE standard 1044-2009 (*IEEE Computer Society, 2010*)):

- errors (behaviour, that is different from what is specified in software requirements specifications or unexpected crashes),
- inconsistencies (not really errors, but disturbing quality failures—e.g., inappropriate input checks, displaying values in inaccurate measures, e.g., the length of a run in meters instead of kilometers, etc.) and
- imperfections (mostly visual quality failures, e.g., using wrong colors, screen appearing incorrect when rotated, etc.).

We preserved this classification also in Figs. 7 and 8. Please note that all failures were addressed (fixed or closed as false failures) during the development. This is why the charts in Figs. 7 and 8 does not capture the frequency of closing failures. The testing efforts were the same for both platforms, so the results are comparable.

As chart in Figure 7 shows, we can observe an almost constant rise of reported failures (5.5 per week; 6.4 if we remove the first 4 weeks of prototyping) in the case of Android development. During the introduction of the SPL approach, the failure number rose from 160

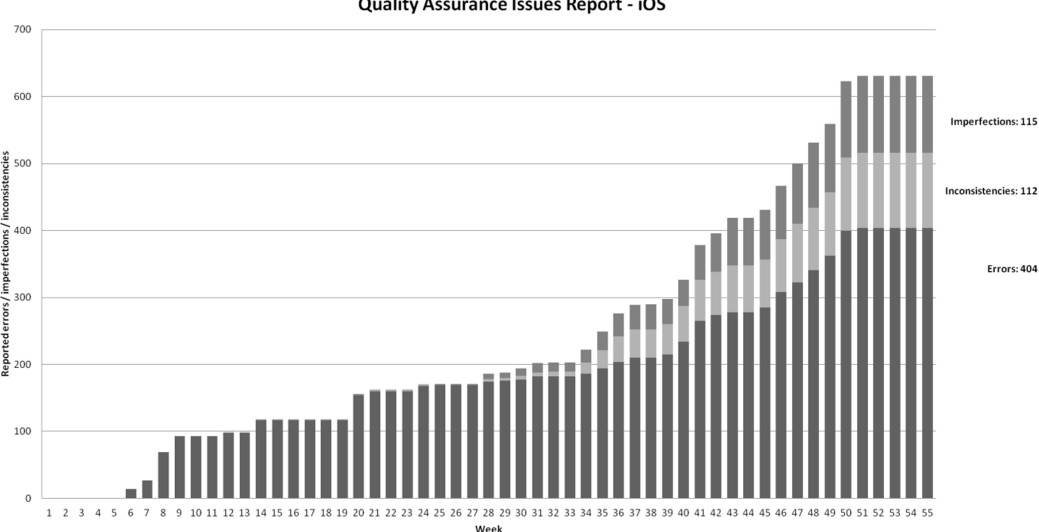

**Figure 8 Cumulative reported failures during time on iOS projects.**

to 240, which is 11.4 per week. In the third period, the final failure count rose from 240 to 405, constant failures per week rate in this period is 9.1. The final failure percentage, according to classifications, is as following: 64% errors, 16% inconsistencies, 20% imperfections.

Similar situation is reported in Fig. 8 for the case of reported failures in iOS development (6.5 per week in first period; 7.8 if we remove the first 4 weeks). During the introduction of the multi-edition approach, the failure number rose from 195 to 280, which is 12 per week. In the third period, the final total failure count rose from 280 to 631, the constant failures per week rate is 25. We can see how the weekly failure rate in this period is not linear. 25 is the average rate during stable multi-edition development. If we split this period in half, the first half has a rate of 18, while the second half is 33. The final failure percentage is 64% errors, 18% inconsistencies, 18% imperfections.

## Post-development survey

After the development, we finalized gathering empirical data to execute a post-development survey among developers. The answers were as follows:

- The development was time framed to 8 h per day. We asked developers about their assessment with regard to how many hours they spent daily on development (allowing for the fact that they might have done extra work at home). iOS developers reported that they spent 9 h a day with the project, Android developers reported working on average 8.7 h daily.
- We asked developers how they liked the implemented mechanism of managing several editions. On a scale of 1–5 (1—do not like it at all; 3—neutral opinion; 5—extremely like it) they responded: iOS developers: 2.7, Android developers: 4.7.
- We asked developers about the fear they had in cases where they would have to change some random code in a project—did they worry that they would cause some failures

with it? On a scale of 1–5 (1—no fear at all; 5—very afraid) they responded: iOS developers: 3.3, Android developers: 1.7.

- We asked developers how they would welcome the need to establish a new edition of a mobile application. On a scale of 1–5 (1—no problem; 5—possible only with high investment) they responded: iOS developers: 4, Android developers: 2.

Finally, developers had opportunity to express their positive and negative experience with managing several editions. The answers include:

- An Android developer highlighted an event as a positive experience. The requirement was stated for the fully functional Pro edition that could be available as a "Demo" edition. Based on Pro, the Demo edition included a special logo and functionality for expiration within 1 month with the possibility of buying the Pro edition on the Google Play market. The edition was prepared and turned over to production, as claimed, in just 3 h.
- As a negative aspect, an iOS developer highlighted fast development, and a lot of requirements changing the already implemented functionalities. Changing/maintaining the code was problematic, he explained.
- An iOS developer suggested, that instead of having many editions of the same applications, it would be better (in terms of development) to have several smaller applications with limited functionalities.

## DISCUSSION

In this chapter we will provide answers to the research questions presented in the "Research method" chapter:

- **RQ1**: Is the source code size comparable between Android and iOS applications, while having the same set of requirements?
- **RQ2**: What are the impacts of introducing a software product lines (SPL) approach to a mobile development?

  - **RQ2.1**: What are the impacts of the SPL approach to application growth?
  - **RQ2.2**: What are the impacts of the SPL approach to application quality?
  - **RQ2.3**: What are the impacts of the SPL approach to developers' confidence in frequent releases?

Based on empirical data from the previous chapter, we will also provide further discussions on the implications that arise from the data. Based on the gathered data, let us answer the fundamental research questions.

### RQ1: is the source code size comparable between Android and iOS applications, while having the same set of requirements?

The final codebase for Android products included 47,259 Java LOC and 32,828 XML LOC; 390 classes (see Figs. 5 and 6). The final codebase for iOS products included 50,739 Objective LOC and 22,744 XML LOC; 290 classes (see Figs. 5 and 6).

Based on the CLOC tool, the final iOS codebase was larger by 7% (Java vs. Objective C); the combined (XML + programing language) LOC is smaller in the case of iOS by 8%. With regard to the LLOC measure (Logical LOC), the final Android LLOC was larger by 7% (see Fig. 5), based on the ProjectCodeMeter tool. The class count was larger in the final Android codebase by 26% (see Fig. 6).

The complete analyzed iOS codebase includes only the Pro and Free editions. On the other hand, the analyzed Android codebase also includes additional editions (Alpha, Test, Demo, BB Pro, BB Free). This is why the Android codebase actually includes 10 more functionalities (see Table 1) compared to iOS codebase functionalities (53 functionalities in Free and Pro), which is 16%. Taking this into account, the effective (normalized to functionalities) LOC difference between iOS and Android is 9% and the LLOC difference is 8% (iOS is higher). The effective class count (normalized to functionalities) difference is 14% (Android is higher). The class difference is easily justified: the Android codebase includes SPL architecture, which is rich in terms of classes. There are also 13 alternative functionalities, where the implementation heavily depends on design patterns and inheritance (which results in introducing new classes—see chapter 4).

To compare product sizes, regardless of SPL approach, we can easily compare codebases at the point of releasing a last single-edition product. This was done in week 33 for Android and week 37 for iOS (see Table 3). Android LOC at that point was 49,664, while iOS LOC was 50,418. The difference is as low as 1.5%. The classes count for Android was 277, while the iOS codebase had 238 classes. Please note, that the difference is 14%, which is the same as the final difference, normalized to functionalities.

**Based on the presented calculations, we can confidently claim, that having the same set of functionalities, implemented using our project boundaries and rules would result in comparable codebase size for both Android and iOS.** This is how we answer the research question 1 as positive.

## RQ2.1: what are the impacts of the SPL approach to application growth?

Observing the data, presented in Fig. 5, we can see that LOC is rising at an almost constant rate of circa 1,400 LOC (pure code, blanks and comments emitted) per week for both the Android and iOS projects. If we analyze the LOC per week coefficient at release weeks or before and after introducing more editions, the data does not show any significant change in LOC velocity. This clearly indicates, that development teams continued to invest unchanged development efforts.

Based on the data presented in Fig. 6 we can see an almost constant coefficient classes/week value of 6. However, in the case of Android, the classes introduction velocity rises to 9 per week after introducing the SPL approach. On the other hand, in the case of iOS, the several edition approach resulted in the dropping of some classes (21).

Implications of quantity metrics are as follows: the developers output in terms of LOC is obviously not affected by multi-edition development. Since the development was functionality-based, it means, that the only driver of LOC velocity were functionalities. However, the structure of the created program code changes if SPL is applied (148 Java

LOC per class before SPL, 98 Java LOC per class with the SPL approach in our case). On the other hand, we can observe even more Objective C LOC per class after ad-hoc multi-edition development in the iOS project. From 153 Objective C LOC per class before multi-edition development, to as high as 189 LOC per class (in week 50) and the final coefficient is 170 LOC per class. Improvement in terms of LOC per class in the iOS project in the last 5 weeks is a result of refactoring. Positive effect on code structure after employing SPL approach is also captured in the CSM measurement (see Table 3).

Based on the presented calculations we can conclude that multi-edition development does not affect development output velocity in terms of LOC, but it does affect the product static structure: SPL-based development in a positive way, while ad-hoc multi-edition development in a rather negative manner.

However, the SPL approach largely affects velocity in terms of functionality and released editions. Based on Tables 1 and 2, we can see that the delivered functionalities and editions per given time frame were comparable in the single-edition period for both Android and iOS development teams. A total of 33 functionalities in single-edition software were delivered in 39 weeks by the iOS development team and in 37 weeks by the Android development team; which is 0.9 functionality per week.

With the same amount of effort input (see post-development survey), this translates into 1.7 new functionalities per week in 2 editions for the iOS development team; and 2.2 new functionalities per week and 1 adopted functionality per week across 7 different editions. While combining public releases with available editions (see Fig. 6), we can see that after multi-edition development, the Android development team delivered 13 (0.93 per week) new editions and versions of software, while iOS delivered 6 (1 per 2 weeks).

Based on presented data, we can answer research question 2.1 as follows. **In our case, SPL approach results in 126% higher functionality-based velocity (or as much as 240% higher, compared to single-edition development). At the same time, the SPL approach enabled developers to adopt an additional 100% of new functionalities across several editions with the same effort level.**

## RQ2.2: what are the impacts of the SPL approach to application quality?

Internal code quality comparison of the products in terms of single-edition development vs. multi-edition development and Android development vs. iOS development is shown in Table 3. In the case of iOS, single-edition and multi-edition development do not change Objective C code quality, which is an expected observation (managing editions was done mainly with compiler directives—see Chapter 4). Modularity improves by 6%, logic density is affected by 5% in a negative manner. The commenting ratio stays the same.

In the case of the Android single-edition and multi-edition development, Java code quality also did not change. Modularity changed by 5% (towards fragmented code), which is expected as a result of SPL implementation (see Chapter 4). The LLOC to capture certain logic increased by 32% (from 45 to 66), which also makes sense: SPL-related

(architecture, optional and alternative functionalities management) code does not capture a lot of business logic. However, logic density is, even after being changed a lot, lower by 19%, compared to Objective C code.

Data shows, that internal code quality did not change importantly. This is a reasonable outcome: the same development skills and effort was used during the whole project. SPL-enabled multi-edition approach affects code in terms of a lot of additional code fragmentation, while the ad-hoc approach does not have this effect. Which is also expected result.

However, Figs. 7 and 8 demonstrate SPL impact on external quality. As implied from the charts showed on Figs. 7 and 8 there are three distinctive periods during the project: single-edition development (weeks 1–33), first multi-edition intensive development (weeks 33–37), and stable multi-edition development.

As explained in "Product Quality" and shown in Fig. 7, in Android application, the constant failures per week is dropped from 11.4 to 9.1 in SPL-enabled period—resulting in 405 failures found throughout the development project.

An important observation from Fig. 7 (Android failures) is this: the constant weekly failure rate in the single-edition development was followed by a rise in failures during the introduction of the SPL approach. During the stable SPL approach-enabled multi-edition development, the weekly reported failure rate is linear and stable again. The rate rose from 6.4 to 9.1 (30%), which is reasonable when considering not only additional functionalities, but also 6 additional editions (from an external point of view completely separate products) with 10 edition-specific and 13 edition-adopted functionalities. The failure increase is significantly smaller than feature and editions increases. Failure per edition on a weekly basis in the last period is as low as 1.3.

As explained in "Product Quality" and shown in Fig. 8, in iOS application, the constant failures per week is elevated from 7.8 to 12 in multi-edition period—resulting in 631 failures found throughout the development project. An important observation from Fig. 8 (iOS failures) is that constant weekly failure rate in single-edition development is followed by an increase in failures during the introduction of the SPL approach. In the period of stable multi-edition development, the weekly failure rate is not linear. The average weekly rate in this period, rose by a factor of 3.2 (from 7.8 to 25). This is, using ad-hoc multi edition development, more failures per edition (25 per week for two editions—12.5 failures per week per edition) than in the single-edition development (7.8 per editions per week). The ad-hoc multi edition approach failed in terms of rising or at least maintaining an external quality level.

Based on presented data, let us answer the research question 2.2. **In our case, internal quality is not affected by introducing SPL approach.** Since SPL approach promotes faster development (see chapter 6.2) and failure rate stays the same, number of failures per functionality drops. **This is how SPL approach enhances external quality. However, using non-SPL multiple edition development results in our case in reduced external quality (reported failures rose by factor 3.2).**

### RQ2.3: what are the impacts of the SPL approach to developers' confidence in frequent releases?

Both the post-development survey and post-development data analysis revealed that developers are more confident in using the SPL approach than using a multi-edition approach, designed and crafted on their own. In the case of SPL-supported multi-edition development, developers created complex software architecture and maintained it while considering strict rules of reuse. They did not have an opportunity to create any shortcut or workaround solution to their approach, as the other development team had. Multi-edition development without an SPL approach relied on developer-chosen and freely implemented approaches, such as compiler directives, runtime conditions, etc.

Therefore, the post-development survey revealed that developers in the SPL-supported development team liked the multi-edition development approach more: 4.7/5, while on the other hand only 2.7/5 in the case of iOS developers. The SPL-supported development team was also more confident in maintaining and developing new code in production software (fear of introducing failures while changing code is as low as 1.7/5 in case of Android developers and as high as 3.3/5 in the case of iOS developers). As a consequence, iOS developers would not be happy with demands for a new edition of the application (4.0/5), while Android developers would welcome it (2.0/5).

Although surprising, we believe that the survey results have a foundation in source code organization: in the case of the SPL approach, the source code of different editions is physically separated. Common functionalities are reused with merging separate source code projects into final products. Therefore, it is clear that developers can be sure that certain new code or code changes will not be included in a certain product edition. On the other hand, ad-hoc multi-edition development has a common code base, where all source code is processed when creating any edition of the final product. Software components are therefore more coupled and less cohesive. As a result, as proven also by the post-development survey, it makes developers not so confident in creating new code or changing existing code.

Higher developer confidence in creating new source code and maintaining existing code was also shown in the post-development data analysis. The failure rate is importantly higher when the SPL approach is not used. The failure introduction velocity is also not linear. It shows how fixing failures in some cases results in introducing new ones with the ad-hoc multi-edition approach. Which finally answers our research question 2.3: **the SPL approach have a positive impact on developers' confidence in delivering releases (and new functionalities as a consequence) frequently.**

### Key findings and limitations

Based on presented results, visualizations and in-depth analysis, let us summarize the main outcomes while answering research questions. Using our research setup, methods and results, we showed that:

- Having the same set of functionalities would result in comparable codebase size for both Android and iOS;

- The SPL approach results in 126% higher functionality-based velocity (240% higher, compared to single-edition development);
- The SPL approach enabled developers to adopt an additional 100% of new functionalities across several editions with the same effort level;
- Internal quality is not affected by introducing SPL approach;
- SPL approach enhances external quality.
- Managing several editions using non-SPL approach would reduce external quality.
- The SPL approach have a positive impact on developers' confidence in delivering new functionalities and releases frequently.

The presented research method, results and discussion also include limitations which represent possible threats to validity.

In ideal world, one would design presented experiment in terms of developing the same application for the same platform twice—with and without a SPL approach to manage several editions. This is how it would be possible to eliminate every doubt, caused by developing for two different operating systems with two different programing languages. However, we created the research method, gather results and interpret them in mind to minimize any doubt in our conclusions. The main measure to address this threat lies in answering research question RQ1. In addition to this, we believe that while we scientifically investigated large, real-world industry project, validity of the results is higher than any laboratory-set experimenting environment.

A question also remains, whether we would end with the same results and conclusion, if in week 33 iOS development team would chose to implement SPL and Android developers would go on with manual approach to manage multiple editions. We addressed this threat by monitoring internal quality of source code and invested effort in terms of delivered LOC. Since internal quality and LOC velocity remains the same after week 33 for both development teams, this indicates that the only difference between products lies in the SPL-enabled architecture in Android application. During the research we have used LOC metric carefully and systematically in both development teams in order to minimize the risk that comes with possible disconnection between LOC metric and delivered functionalities.

## CONCLUSIONS

This paper highlighted our research during two development teams that created production-ready mobile application families for two different platforms (Android, iOS). They shares the same set of functionalities and were done with the same effort input. Both families share 85% of their common functionalities. Other functionalities are optional or alternative. One product family was developed with the SPL approach to manage reuse, while other product family was developed with more traditional and ad-hoc reuse techniques (single codebase, no special architecture to manage variabilities, employing compiler directives, etc.). As shown in this paper, this was the only difference in approach between the development of two functionally equal software families.

The results of this research show that two product families not only share the same set of functionalities and effort rate, but, surprisingly, product growth and end products size were also the same. Comparing multi-edition with single-edition development, we showed that development velocity, in terms of code quantity, did not change. However, code structure improved after introducing the SPL approach. The SPL approach also had positive implications for feature-based output: after the introduction of the SPL approach, functionality output per week increased by 240%, which is 124% higher than cases of iOS development with non-SPL approach to multi-edition development. At the same time, the SPL approach employment resulted in 6 new editions, while the non-SPL approach resulted in only one.

An important aspect, directly impacted by the SPL approach, is also software quality. With the SPL approach, the failure introduction velocity did not increase, but remained linear. After dividing failures across all editions, failure rates fell. On the other hand, we showed how the failure introduction rate in the case of the non-SPL approach not only increased but was also not linear anymore. In our case, the failure count, divided by editions, is even higher than in the single-edition development. The derived statement from this observation is that doing multi-edition development without an SPL approach is something that will have a very bad impact in terms of software quality. On the other hand, the SPL approach has a very positive impact on software quality.

Our research revealed that the SPL approach enables development teams to produce more functionalities in several product editions with the same effort as they would use while working on a single-edition product. Not only productivity, but even more importantly, software quality, rises. Developers' confidence in the maintenance and developing of new code is also higher, when using the SPL-based product structure.

Our finding, that is even more taught full is, that when development teams have to manage multiple editions, doing it manually can have severe consequences in terms of drop in quality and worse developers' confidence while introducing new functionalities or upgrading existing ones.

### Funding
Financial support was received from the Slovenian Research Agency (research core funding No. P2-0057). The funders had no role in study design, data collection and analysis, decision to publish, or preparation of the manuscript.

### Grant Disclosures
The following grant information was disclosed by the authors:
Slovenian Research Agency: P2-0057.

### Competing Interests
The authors declare that they have no competing interests.

## Author Contributions

- Luka Pavlič conceived and designed the experiments, performed the experiments, analyzed the data, performed the computation work, prepared figures and/or tables, authored or reviewed drafts of the paper, and approved the final draft.
- Tina Beranič conceived and designed the experiments, performed the experiments, analyzed the data, performed the computation work, prepared figures and/or tables, authored or reviewed drafts of the paper, and approved the final draft.
- Marjan Heričko conceived and designed the experiments, performed the experiments, authored or reviewed drafts of the paper, and approved the final draft.

## Data Availability

Data is available at GitHub:

https://github.com/luka-pavlic/SPL.

Data is also available at the University of Maribor:

http://spl.informatika.uni-mb.si.

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
