# Peer review of "A product quality impacts of a mobile software product line: an empirical study"

_PeerJ Computer Science, doi:10.7717/peerj-cs.434_

## Round 0.1 · original submission · Major Revisions

The reviewers completed their evaluation. Based on their comments, we cannot accept your paper for publication without major revisions.

In the revised paper, please address all the reviewers' comments.

Please include more data about the process followed to develop the software product line and justify choosing the RQs as the metrics to measure the benefits of such SPL. Furthermore, please add more details in your experiment's description because now it is impossible to conduct a replication study due to the lack of information provided. The contribution of this work should also be clarified.

Reviewer 1 ·

Basic reporting

The paper is technically sound and well-written. There is a typo in section four’s title “An project’s…” (line 239).
The background and literature review provided is wide and enough to understand the context in which this work is framed. Also, the paper is well organized.

Experimental design

The experiment’s design is interesting, however, I think a section for the SPL development process (the development of core assets, variability points, etc.) with deeper details would be beneficial to understand both the approach and the experiment’s results.

Validity of the findings

The results are well discussed and all the data supporting them provided, however, I have a few questions that need to be justified in the paper:

- Why the Android app as a SPL? Could the findings be different if the SPL was developed for the iOS app? Why?
- Is there an estimate on how much time, resources and effort the change of methodology from manual development to a SPL consumed?

Additional comments

The paper provides an interesting experiment regarding the introduction of a SPL within the mobile applications’ domain. A visual example of the workflow followed regarding the application engineering process would be appreciated. I would like to know if the authors plan to carry out usability experiments with both apps to test if the SPL introduction had any impact for the user.

Reviewer 2 ·

Basic reporting

No comment.

Experimental design

My main concerns about this paper are in the experimental design:

1) Why the decision of making a comparison using two different operating systems and programming languages? The results may be biased by the important differences between these two systems. For me, it would have more sense to develop the SPL-based and the non-SPL versions of the system in the same operating system, e.g. Android. In this case, it would be much easier to compare the results and to identify the benefits of using an SPL. Even if the Android project were using both approaches in different periods of the project the results are difficult to be compared.

2) I do not completely understand how in RQ1 the lines of code can be comparable. Even if the application is the same, the operating system and the programming language are different and therefore this has to be also taken into account in order to do the comparison. Once again this RQ would have much more sense if the comparison were between the same applications, being developed with and without a SPL approach.

3) There is not enough information in the paper about the system under development. The description of the system is too informal. A more formal and detailed description of the system requirements, design, etc... would be needed.

4) Figure 2 is not a representation of an SPL. Which notation are you using? Which is the mandatory/optional part? Which are the constraints? I mean an SPL implies certain phases that are not considered in the paper or at least not documented. For instance, there is a differentiation between the Problem Space and the Solution Space. In the Problem Space the variability is modeled using traditional a feature model. This feature model includes optional features, mandatory features, groups and constraints. From this feature model, the number of different valid products can be counted and configurations for each product are automatically generated. In the solution space the code of each feature is developed and connected somehow with the features in the feature model. In case you are using another approach, this should be well documented in the paper.

Validity of the findings

The findings would need to be better organized and better explained.

Organization. There are too much textual information when it would be more readable to have several tables organizing the information.

Explanation. I do not completely understand how the reasoning about the LoC reveals that there is a benefit of using an SPL. I have the same impression with the answer to the other RQs. I do not completely see how the analyzed results help to conclude that the use of a SPL has benefits.

Additional comments

I think the motivation for this paper is very interesting. However, I am not convinced about the experimental design and the findings for this work (as indicated in my previous comments).

In summary;
- There is a lack of information about how the SPL has been defined. What does authors understand by applying an SPL to their project? I think that this should be clarified because Figures 2 and 3 are not a representation of a traditional SPL. Thus, authors need to make clearer the approach they have followed to define their software product line.
- The criteria used to quantify the benefits of using a SPL need to be reviewed. For instance, in some implementations of a SPL the number of lines of code can be considerably increased but this is because the automatic generation of code for the different products introduces some benefits that compensate for the increment in the lines of code. Since here it is not clear how the SPL has been developed it is very difficult to reason about the benefit regarding the number of line of code. The same can be said for the other RQs.

---

## Round 0.2 · Minor Revisions

Thanks for your work on improving the paper.

Please address carefully Reviewer 2's comments in order to get your work accepted.

Reviewer 1 ·

Basic reporting

The paper shows an interesting approach to study the impact of the introduction of the SPL paradigm in the development process of a mobile app. The work is organized and well-written.

Experimental design

With the revised manuscript, the experimental design is more clear.

Validity of the findings

The discussion and limitations are detailed and provide clear explanations regarding the obtained results.

Additional comments

Thank you for the detailed changes and explanations on the revised manuscript. The paper has been clearly improved.

Reviewer 2 ·

Basic reporting

No comment

Experimental design

I would like to thank the authors for working on all the comments in my previous review. In this review, I went over all my previous comments and the author's response and I will comment on them.

Response #01 and #06. Comparison between and Android and IoS applications.
The response given by the authors to this comment made me understand much better the motivation and value of the work. However, I think that the authors explained this motivation much better in their responses #01 and #06 than in the paper itself. In the paper, this description is split between section 3 and section 4 and I think it should be all explained in the same section.

Recommendation to improve the paper: check again how the work is motivated in the paper and try to move some of the discussion in responses #01 and #06 to the paper. The information is in the paper but slightly reduced and organized and explained in a different way and I think it is much more clear in their response. The research method discusses the two mobile applications but they are not introduced yet (they are introduced at the beginning of section 4). I would move the description before section 4.1 to section 3. The reader needs to understand the context of what has been developed before describing the research method.

Response #7. Line of Codes comparison.
I kind of understand why the authors monitor the line of codes in both projects, but still do not completely understand how the number of line of codes, by themselves, can provide authors with such amount of information: whether products are comparable, that functionalities are the same, that using an SPL does not imply additional changes. For instance, it may easily happen, based on my experience, that for the same functionalities the number of lines of codes may simply be completely different and that doesn´t mean that the two systems offer different functionalities.

Response #9. SPL representation.
Well, I have to say that I disagree that the SPL is "something conceptual and does not imply an implementation". It is not only a concept, and it is not only to have a set of components that are added/removed from a system. Software Product Line Engineering is much more than this and provides clear phases of how to develop a software product line and clear artifacts that have to be defined/implemented in each phase, etc.
Said that I understand that authors have better explain now the mechanism that Android developers used to implement the variation points in the SPL.

Validity of the findings

Comment to Response #10. Better organize the findings.
I was not aware that the journal requested more text than tables or graphical elements. I any case, in my opinion, it is important that the results are presented and discussed in a format that helps to understand the conclusions. I still consider that the discussion about the obtained result is difficult to follow and it is very easy to be lost in the details.

Recommendation: Better explain some sections. For instance, in sections 5.4 and 5.5 authors merely refer to the table or figures, but an explanation of the results and the meaning of that results are missed. In general, when a table or figure is included this improves the readability so I completely agree with them, but they need to be referenced and explained in the text.

Additional comments

- Pre- and post questionnaires and their responses should be included as Annexes.

- I still have problems understanding the results of your experiments.

---

## Round 0.3 · accepted · Accept

The manuscript has been revised and has been accepted. Congratulations!